# Factors Exacerbating Clinical Symptoms and CT Findings in Patients with Medication-Related Osteonecrosis of the Jaw Receiving Conservative Therapy: A Multicenter Retrospective Study of 53 Cases

**DOI:** 10.3390/ijerph19137854

**Published:** 2022-06-26

**Authors:** Yuka Kojima, Sakiko Soutome, Mitsunobu Otsuru, Saki Hayashida, Yuki Sakamoto, Shunsuke Sawada, Masahiro Umeda

**Affiliations:** 1Department of Dentistry and Oral Surgery, Kansai Medical University, Hirakata 573-1191, Japan; sawadash@hirakata.kmu.ac.jp; 2Department of Oral Health, Graduate School of Biomedical Sciences, Nagasaki University, Nagasaki 852-8588, Japan; sakiko@nagasaki-u.ac.jp; 3Department of Clinical Oral Oncology, Graduate School of Biomedical Sciences, Nagasaki University, Nagasaki 852-8588, Japan; ootsuru@nagasaki-u.ac.jp (M.O.); sakihaya@nagasaki-u.ac.jp (S.H.); mumeda@nagasaki-u.ac.jp (M.U.); 4Department of Dentistry and Oral Surgery, Kansai Medical University Medical Center, Moriguchi 570-8507, Japan; s.yukioutdoor@gmail.com

**Keywords:** medication-related osteonecrosis of the jaw, conservative treatment, periosteal reaction, exacerbation, treatment outcome

## Abstract

Recent reports have shown that better treatment outcomes are obtained with surgical therapy in patients with medication-related osteonecrosis of the jaw (MRONJ) than with conservative therapy. However, conservative treatment is selected due to factors such as old age and poor general condition. Conservative therapy aims to relieve symptoms and prevent lesion exacerbation; however, the lesion may expand rapidly in some cases. This study investigated the clinical and imaging findings of 53 MRONJ patients undergoing conservative therapy, and the changes in the clinical findings and the lesion enlargement on CT imaging were examined. Improved clinical findings and no worsening of the imaging findings were considered overall comprehensive treatment successes. Among the 53 patients, the clinical symptoms disappeared or improved in 15 patients, whereas they worsened in 6. In contrast, osteolytic lesion enlargement occurred in 17 patients. The comprehensive treatment outcome of conservative therapy was successful in 12 patients and unsuccessful in 41 patients. The periosteal reaction was significantly correlated with poor comprehensive treatment outcomes (*p* = 0.038). MRONJ lesions may advance, even if they appear to improve clinically while undergoing conservative treatments. Patients, especially those with periosteal reactions, must be closely followed up with CT examinations, regardless of the clinical findings.

## 1. Introduction

Antiresorptive agents, such as bisphosphonate and denosumab, are widely used to prevent fracture in patients with osteoporosis or to treat skeletal-related events in those with bone metastasis of malignant tumors or multiple myeloma. However, these drugs sometimes cause medication-related osteonecrosis of the jaw (MRONJ). In most stage 1 and 2 patients, conservative therapy was recommended as the first-line therapy by the American Association of Oral and Maxillofacial Surgeons (AAOMS) Position Paper of 2014 [1], the Japanese Allied Committee on Osteonecrosis of the Jaw Position Paper of 2017 [2], and the MASCC/ISOO/ASCO Clinical Practice Guideline 2019 [3]. The AAOMS Position Paper was revised in 2022 [4], and some modifications were made to the treatment strategy. It describes how both conservative and surgical treatments are acceptable for all stages of MRONJ, depending on the patient’s situation. In contrast, recent reports have shown that the treatment outcome of those who undergo surgery is significantly better than that of those who undergo conservative therapy [5,6,7,8]. In addition, a multicenter retrospective study of 361 patients with MRONJ showed that surgical therapy yielded significantly better outcomes than conservative therapy by propensity score matching analysis [9]. 

Surgery may not be selected for elderly patients with osteoporosis or patients with cancer and who are in poor general condition; minimally invasive surgical procedures or conservative therapy is preferred in such cases. Conservative therapy includes local irrigation, antibacterial gargling, and the administration of antibiotics. After a long period of conservative therapy, the sequestrum may separate, and healing may be achieved by removing it; however, in many cases, the goal of conservative therapy is to prevent the progression of the lesions and relieve clinical symptoms. Rapid exacerbation of the MRONJ lesion may occur occasionally during conservative treatment. This retrospective study aimed to examine the clinical and imaging outcomes of patients with MRONJ undergoing conservative therapy and examine the factors related to the outcomes of conservative treatment.

## 2. Materials and Methods

### 2.1. Patients and Study Design

This is a retrospective observational study. The study enrolled patients with MRONJ who underwent conservative treatment for 6 months or longer at the Kansai Medical University Hospital or the Nagasaki University Hospital, Japan, between 2011 and 2019. Since this is a retrospective observational study, the sample size was not calculated, and all cases during the target period were enrolled. Patients prior to 2011 were not included in the study because sufficient information was not digitized and stored. In addition, patients who were followed up for less than 6 months were excluded.

### 2.2. Conservative Treatment

Conservative treatment includes the following procedures: oral hygiene guidance, gargling with antibacterial mouthwash, local lavage, and administration of oral antibiotics such as amoxicillin, clindamycin, and sitafloxacin when the infectious symptoms are strong. Further, removal of mobile segments of bony sequestrum and extraction of symptomatic teeth within exposed necrotic bone are also considered in those undergoing conservative treatment.

### 2.3. Variables

The variables examined in the study were sex, age, jaw (upper or lower jaw), site (anterior, posterior, or anteroposterior), MRONJ stage, type of antiresorptive agent (bisphosphonate or denosumab), primary disease (osteoporosis or malignant tumor), administration period of antiresorptive agent (<4 years or ≥4 years) [1], drug holiday during treatment, administration of corticosteroid, diabetes, leukocyte count, serum level of albumin and creatinine before treatment, separation of the sequestrum, the extent of the osteolytic lesion, and periosteal reaction. MRONJ stage was classified according to the AAOMS 2014 Position Paper [1].

The site was defined as “anterior” from the incisor to the canine region, “posterior” from the premolar region, and “anteroposterior” included both sites. The separation of the sequestrum, the extent of the osteolytic lesion, and periosteal reaction were determined using computed tomography (CT) in bone mode. Regarding the sequestrum separation, those showing a radiolucent area between the necrotic bone and the surrounding bone in almost the entire area were classified as “with separation of sequestrum,” whereas the remaining were classified as “without separation of sequestrum” (Figure 1A). The extent of the osteolytic lesion was classified into “above the mandibular canal” and “including the mandibular canal” (Figure 1B,C). The lesions occurring in the anterior mandible were classified based on the hypothetical line of mandibular canal height. The periosteal reaction was classified into two types: with and without (Figure 1D).

### 2.4. Outcome

The outcome is defined by the clinical and imaging findings. The clinical findings include infection findings such as bone exposure, swelling, pain, redness, pus discharge, and cutaneous fistula at the first visit and during follow-up. The patients in whom all clinical findings had disappeared were classified as “healing”, those with improved clinical findings were classified as “improved”, those with no change in the clinical findings were classified as “no change”, and those with deterioration in the clinical findings were classified as “worsening”. The timing was also recorded for patients with changes in clinical symptoms. Regarding the imaging findings, the osteolytic lesions in the CT examinations taken during observation were classified as “disappearance” for those in whom the osteolytic lesions had disappeared, “reduction” for those in whom the osteolytic lesions had reduced, “no change” for those in whom the osteolytic lesions did not change, and “enlargement” for those in whom the osteolysis had increased. For patients who underwent CT imaging three or more times, the time at which there was a change in CT findings was recorded. The final CT time was recorded when there was no change. Furthermore, since the purpose of conservative therapy for MRONJ is to prevent the growth of the lesions and relieve the symptoms, patients with no clinical symptoms or improved clinical symptoms and no increase in osteolysis on CT findings were defined as “comprehensive treatment success”, and patients with no improvement in clinical symptoms or increased osteolysis on CT findings were defined as “comprehensive treatment failure”.

### 2.5. Statistical Analysis

The relationship between each variable and the outcome of the clinical or imaging findings was analyzed by one-way ANOVA or Fisher’s exact test. Univariate analysis was performed, and multivariate analysis was attempted after considering the number of events and each *p*-value. All statistical analyses were performed using SPSS software (version 26.0, Japan IBM Co., Tokyo, Japan). Two-tailed *p*-values < 0.05 were considered statistically significant.

## 3. Results

### 3.1. Patient Characteristics

The study enrolled 53 patients with MRONJ. Surgical treatment is the first-line treatment for patients with all stages of MRONJ at both hospitals; 259 patients underwent surgery.

The patients consisted of 18 males and 35 females with an average age of 74.9 years. The site of MRONJ was the upper jaw in 10 patients and the lower jaw in 43, and 20 patients had osteoporosis, while 33 had a malignant tumor. In 36 patients (67.9%), the antiresorptive agent was discontinued during conservative therapy. The summary of the patients is shown in Table 1.

### 3.2. The Outcome of Clinical and Imaging Findings and Comprehensive Treatment Outcome

Among the 53 patients, the clinical symptoms of 15 (28.3%) disappeared or improved, while worsening was observed in 6 (11.3%). In contrast, the image findings did not disappear in any patients, and only two patients (3.8%) showed a reduction in the osteolytic area. Enlargement of the osteolytic lesion occurred in 17 (32.1%) patients (Table 2). These findings indicate that it is not uncommon for lesions to advance, even if they appear to improve clinically (Figure 2).

The comprehensive treatment outcome of conservative therapy was successful in 12 (22.6%) patients and unsuccessful in 41 patients (77.4%).

### 3.3. Factors Related to Clinical Findings and Imaging Findings

Table 3 shows the relationship between each variable and the clinical symptoms as well as the imaging findings by univariate analysis. An improvement in the clinical symptoms was observed in five out of nine patients (55.6%) in stage 1, 8 out of 30 (26.7%) in stage 2, and 2 out of 20 (10%) in stage 3. The clinical symptom improvement rate for conservative therapy decreased as the stage progressed; however, no significant difference was observed (*p* = 0.096). The underlying disease did not affect the clinical symptom improvement rate (*p* = 1.000). The clinical symptom improvement rate was slightly higher in cases with separation of the sequestrum (*p* = 0.076) and slightly lower in cases with periosteal reaction (*p* = 0.055); however, there was no statistically significant difference in either case. 

Subsequently, the factors related to the enlargement of the osteolytic lesion on the CT images were investigated by univariate analysis. The MRONJ stage (*p* = 0.076) and primary disease (*p* = 0.225) were not associated with the enlargement of osteolysis on the CT images. The enlargement of osteolysis was less in cases with separation of the sequestrum; however, there was no significant difference. In 11 out of 19 patients with periosteal reaction, the lesions were increased on the CT images. The worsening rate of the CT findings was significantly higher in 6 out of the 34 patients without periosteal reaction(*p* = 0.005).

Multivariate analysis could not be performed due to the small number of events.

### 3.4. Factors Related to Comprehensive Treatment Outcome

The relationship between each variable and the comprehensive treatment outcome is shown in Table 4. Clinical factors such as age, site, primary disease, type of antiresorptive agent, administration period, and drug holiday were not associated with the outcome; however, patients with stage 3 MRONJ showed significantly worse outcomes (stage 1–2 vs. stage 3: *p* < 0.001). Furthermore, those with a periosteal reaction on CT examination were significantly correlated with poor comprehensive treatment outcomes (*p* = 0.038). 

## 4. Discussion

Surgical treatment has significantly better treatment outcomes compared to conservative treatment for MRONJ [5,6,7,8,9]. However, conservative therapy may have to be selected due to reasons such as old age and the poor general condition of patients. Several investigators have previously recommended conservative treatment for MRONJ. Lerman et al. [10] reported in a retrospective observation of 120 records that a primarily non-surgical approach appears to be a successful management strategy for MRONJ and that 71–80% of patients improved or remained asymptomatic with a median follow-up of 12 months. Marx et al. also stated that in patients with MRONJ with painful exposure of the bone, the effective control rate to a painless state was over 90% with antibiotics administration along with 0.12% chlorhexidine antiseptic mouthwash use [11]. The AAOMS Position Paper 2022 developed a series of treatment algorithms to streamline the evaluation and management strategies for patients with MRONJ. It described that both nonoperative and operative management is acceptable for all stages of the disease [4]. In contrast, Lazarovici et al. [12] reported conservative treatment as the regimen of choice based on sequestrectomy and trimming of the exposed bone with an empiric antibiotic treatment protocol; however, the solutions for decreasing morbidity and poor outcomes of ONJ remain elusive. 

Some researchers have attempted to obtain good treatment results by adding various adjuvant therapies to conservative treatments or conservative surgery. Hyperbaric oxygen therapy (HBO) [13,14]; ozone [15,16,17]; low-level laser therapy (LLLT) [18,19,20]; a combination of laser ablation and LLLT [21,22,23]; leukocyte-rich and platelet-rich fibrin (L-PRF) and its variations [24,25,26,27,28]; teriparatide [29,30]; laser ablation and L-PRF [31]; combined laser ablation, LLLT, and L-PRF [32]; and combined bone morphogenetic protein (BMP)-2 and L-PRF [33] have been reported to be useful as adjuvant therapies to conservative treatments or conservative surgical treatment. However, these studies only had a few cases, and several did not have controls; thus, their evidence remains questionable.

Although various studies have shown that surgical therapy has a high cure rate for MRONJ, there are many elderly patients with osteoporosis and patients with distant metastases and poor systemic disease, and some suggest that minimally invasive conservative therapy should be performed first; surgical therapy should only be considered in cases showing poor progress [1,2,3]. In osteoporosis patients on low-dose antiresorptive agents, if the drug is withdrawn or replaced with another resorption inhibitor that does not cause MRONJ, even conservative therapy may result in the separation of the sequestrum, which can then be removed under local anesthesia for healing [34]. However, while surgery under general anesthesia can be avoided, it often takes a long time to heal, and patients must continue to make outpatient visits during conservative treatment. Even when the sequestrum is separated, the initial extent of necrotic bone is not reduced and may even be enlarged. In our opinion, conservative therapy is not necessarily less invasive than surgical therapy. In contrast, in patients with malignant tumors receiving high-dose antiresorptive agents, withdrawing the drug is difficult given the treatment of the underlying disease, even if conservative therapy is chosen. Unlike osteoporosis, there are no alternatives to bisphosphonate or denosumab in patients with malignancy. The cure rate with conservative therapy in patients receiving high-dose antiresorptive agent therapy is low [9]. The early detection of MRONJ and early surgery can lead to early cure through minimally invasive procedures, such as marginal mandibulectomy. Therefore, we believe that surgical therapy is appropriate as a first-line treatment for patients who can undergo surgery. Nevertheless, there are still some patients for whom surgical therapy is not the first-line treatment of choice due to systemic diseases or patient preference.

The goal of conservative treatment for MRONJ is to prevent the progression of the lesions and relieve clinical symptoms [1,2]. However, the lesions may sometimes grow rapidly during conservative treatment and may subsequently require larger invasive surgery. In this study, there were 15 patients whose clinical findings disappeared or improved; however, only two showed an improvement in their CT findings. Furthermore, although the clinical findings worsened in 6 patients, advanced lesions on CT examination were observed in 17 patients. It should be noted that although the clinical findings appeared to have improved, the lesions may have expanded during conservative therapy. Predicting whether conservative therapy will relieve symptoms or exacerbate the lesions will be of great benefit when choosing a treatment for MRONJ.

The results of this study suggest that patients with a periosteal reaction on CT examination before treatment may experience worsening of the lesions during conservative therapy, although it is difficult to draw a clear conclusion, as the number of events of exacerbation was as small as 6 in the clinical findings and 17 in the CT findings. Periosteal reaction is a phenomenon sometimes seen in chronic osteomyelitis and malignant tumors of the mandible and is considered a type of reactive osteogenesis in the living body [35]. Suei reported that periosteal reaction of the jawbone was observed by CT examination in 15/25 (60%) cases of MRONJ, 2/36 (6%) cases of radiation osteomyelitis, 39/92 (42.4%) cases of suppurative osteomyelitis, and 29/34 (85%) cases of diffuse sclerosing osteomyelitis; however, the clinical significance of periosteal reaction was not mentioned [36]. We previously reported in a multicenter retrospective study on 164 MRONJ surgeries that periosteal reaction was found during preoperative CT examination in 21.3% of cases, and periosteal reaction is associated with a lower cure rate despite surgical treatment [37]. Furthermore, we classified periosteal reaction in MRONJ into three types: attached, gap, and irregular. We also advocated that the complete resection of both the osteolytic areas and the irregular type of periosteal reaction was necessary to obtain complete healing in patients undergoing mandibulectomy [38,39]. We hypothesized that the periosteal reaction in MRONJ is not a reactive phenomenon (e.g., similar to chronic osteomyelitis) but a more infectious, destructive lesion. MRONJ with periosteal reaction suggests a more aggressive lesion and is associated with poor outcomes after surgery. The study found that the presence of periosteal reaction predicts a poor treatment outcome in surgical cases and in patients undergoing conservative treatment.

As mentioned above, conservative therapy does not have a high cure rate for MRONJ; therefore, the prevention of MRONJ is important. It was recommended that patients receiving antiresorptive agent therapy avoid invasive dental procedures such as tooth extractions and that antiresorptive agents be withdrawn before tooth extraction, which resulted in the preservation of the tooth with a source of infection. Recently, Soutome et al. reported a significantly higher incidence of subsequent MRONJ in the presence of teeth with periapical lesions larger than 3 mm, periodontal pockets larger than 4 mm, and local infection symptoms [40]. Since extraction itself is not a risk factor for MRONJ, but local infection present in the tooth to be extracted is, the extraction of such teeth may lead to the prevention of MRONJ development [41].

This study had some limitations. First, the number of cases was small, and no factors related to the results could be found other than the periosteal reaction. Second, since the patients receiving conservative therapy in this study were selected from hospitals that offer surgical treatment as the first-line treatment, there is a selection bias, and it is difficult to generalize the results obtained. Furthermore, since this was a retrospective survey, the timing of postoperative CT examinations may vary, and a fixed follow-up method was not followed. However, this study is the first report showing that periosteal reaction is one of the poor prognostic factors for patients with MRONJ undergoing conservative therapy, and we believe that its significance is great. In the future, we would like to increase the number of cases and conduct further prospective observational studies.

## 5. Conclusions

The clinical and imaging findings of patients with MRONJ treated with conservative therapy were examined, and the factors related to the worsening of the MRONJ lesion were investigated. Even if the clinical findings are improved by conservative therapy, the lesions may be exacerbated on CT images. In patients with MRONJ, especially in those with periosteal reaction, it is necessary to closely follow up with CT examinations regardless of the clinical findings.

## Figures and Tables

**Figure 1 ijerph-19-07854-f001:**
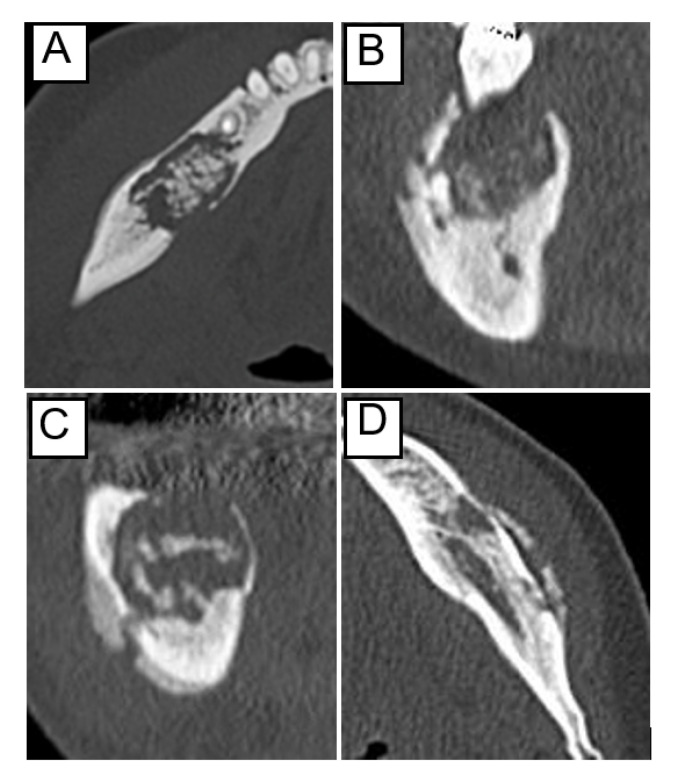
Examples of CT findings before treatment: (**A**) separation of the sequestrum, (**B**) osteolysis above the mandibular canal, (**C**) osteolysis including the mandibular canal, and (**D**) periosteal reaction.

**Figure 2 ijerph-19-07854-f002:**
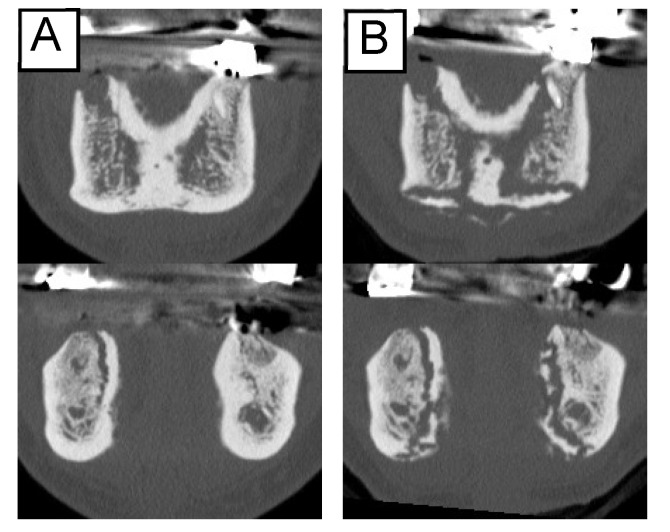
Changes in the imaging findings. The extent of osteolysis increased despite improved clinical findings: (**A**) before treatment; (**B**) nine months later.

**Table 1 ijerph-19-07854-t001:** Patient characteristics.

Variable		Number of Patient/Mean ± SD
Sex	male	18
	female	35
Age	(years)	74.9 ± 11.9
Jaw	upper jaw	10
	lower jaw	43
Site	anterior	5
	posterior	34
	anteroposterior	14
Stage	stage 1	9
	stage 2	30
	stage 3	14
Antiresorptive agent	bisphosphonate	32
	alendronate	11
	zoledronate	18
	minodronate	1
	risedronate	2
	denosumab	16
	both	5
	alendronate + denosumab	1
	zoledronate + denosumab	4
Primary disease	osteoporosis	20
	malignant tumor	33
Administration period of antiresorptive agent	(months)	47.0 ± 33.9
	<4 years	29
	≥4 years	24
Drug holiday during treatment	(−)	12
	(+)	36
	unknown	5
Corticosteroid	(−)	42
	(+)	11
Diabetes	(−)	43
	(+)	10
Leukocytes	(/μ)	6920 ± 2691
Albumin	(g/dL)	3.75 ± 0.567
Creatinine	(mg/dL)	1.06 ± 0.540
Separation of sequestrum	(−)	41
	(+)	12
Osteolytic lesion	above the mandibular canal	34
	including mandibular canal	19
Periosteal reaction	(−)	34
	(+)	19
Observation period	days	729 ± 494
Total		53 patients

**Table 2 ijerph-19-07854-t002:** Treatment outcome.

Treatment Outcome		Number of Patients	Observation (Mean)
Clinical symptoms	Healing	8	249–1447 days (731)
	Improvement	7	149–1163 days (462)
	No change	32	147–1955 days (759)
	Worsening	6	111–1432 days (884)
Osteolytic lesion on CT image	Disappearance	0	-
	Reduction	2	320–644 days (482)
	No change	34	111–1819 days (650)
	Enlargement	17	160–1955 days (917)
Comprehensive treatment outcome	Success	12	149–1447 days (592)
	Failure	41	111–1955 days (770)
Total		53	111–1955 days (729)

**Table 3 ijerph-19-07854-t003:** Factors related to treatment outcome.

Variable		Clinical Symptoms				Osteolytic Lesion of CT Image			
		Number of Patient/Mean ± SD		Odd Ratio	*p*-Value	Number of Patient/Mean ± SD		Odd Ratio	*p*-Value
		Healing/Improvement	No Change/Worsening			Disappearance/Reduction/No Change	Enlargement		
Sex	male	7	11	1.000	0.334	10	8	1.000	0.218
	female	8	27	0.296		26	9	2.311	
Age	(years)	76.1 ± 11.8	74.4 ± 12.0	1.024	0.649	76.1 ± 11.5	72.2 ± 12.6	1.028	0.272
Jaw	upper jaw	2	8	1.000	0.706	8	2	1.000	0.471
	lower jaw	13	30	2.143		28	15	0.467	
Site	anterior	2	3	1.000	0.614	3	2	1.000	0.651
	posterior/anteroposterior	13	35	0.557		33	15	1.457	
Stage	stage 1–2	5	4	1.000	0.096	27	12	1.000	0.776
	stage 3	2	12	0.487		9	5	0.800	
Antiresorptive agent	bisphosphonate	9	23	1.000	1.000	21	11	1.000	0.768
	denosumab or both	6	15	1.100		15	6	1.310	
Primary disease	osteoporosis	6	14	1.000	1.000	16	4	1.000	0.225
	malignant tumor	9	24	1.167		29	13	0.385	
Administration period	(months)	40.9 ± 32.7	49.2 ± 34.4	0.990	0.454	51.1 ± 37.8	38.7 ± 22.5	1.013	0.234
	<4 years	9	20	1.000	0.762	17	12	1.000	0.145
	≥4 years	6	18	0.545		19	5	2.682	
Drug holiday during treatment	(−)	4	8	1.000	0.331	24	12	1.000	0.800
	(+)	11	25	1.000		9	3	1.500	
	unknown	0	5			3	2		
Corticosteroid	(−)	11	31	1.000	0.708	29	13	1.000	0.730
	(+)	4	7	2.083		7	4	0.784	
Diabetes	(−)	12	31	1.000	1.000	30	13	1.000	0.709
	(+)	3	7	0.888		6	4	0.650	
Leukocytes	(/μ)	7433 ± 2491	6728 ± 2775	1.000	0.446	7116 ± 3157	6500 ± 1202	1.000	0.486
Albumin	(g/dL)	3.76 ± 0.520	3.75 ± 0.591	0.696	0.995	3.74 ± 0.523	3.77 ± 0.664	0.917	0.887
Creatinine	(mg/dL)	0.947 ± 0.290	1.11 ± 0.606	1.237	0.383	1.05 ± 0.527	1.10 ± 0.587	0.837	0.769
Separation of sequestrum	(−)	9	32	1.000	0.076	25	16	1.000	0.077
	(+)	6	6	2.727		11	1	7.040	
Osteolytic lesion	localized	9	25	1.000	0.756	23	11	1.000	1.000
	extended	6	13	1.400		13	6	1.036	
Periosteal reaction	(−)	13	21	1.000	0.055	28	6	1.000	0.005
	(+)	2	17	0.431		8	11	0.156	

**Table 4 ijerph-19-07854-t004:** Factors related to success or failure of treatment.

Variable		Comprehensive Treatment Outcome			
		Number of Patient/Mean ± SD		Odd Ratio	*p*-Value
		Success	Failure		
Sex	male	5	13	1.000	0.730
	female	7	28	0.650	
Age	(years)	75.2 ± 12.6	74.8 ± 11.8	1.003	0.922
Jaw	upper jaw	2	8	1.000	1.000
	lower jaw	10	33	1.212	
Site	anterior	2	3	1.000	0.315
	posterior/anteroposterior	10	38	0.395	
Stage	stage 1–2	10	29	1.000	0.480
	stage 3	2	12	0.483	
Antiresorptive agent	bisphosphonate	6	26	1.000	0.507
	denosumab or both	6	15	1.733	
Primary disease	osteoporosis	4	16	1.000	1.000
	malignant tumor	8	25	1.280	
Administration period	(months)	39.6 ± 35.3	49.2 ± 33.6		0.417
	<4 years	7	22	1.000	1.000
	≥4 years	5	19	0.972	
Drug holiday during treatment	(−)	8	28	1.000	0.434
	(+)	4	8	2.063	
	unknown	0	5		
Corticosteroid	(−)	9	33	1.000	0.697
	(+)	3	8	1.375	
Diabetes	(−)	11	32	1.000	0.423
	(+)	1	9	0.323	
Leukocytes	(/μ)	7459 ± 2648	6761 ± 2722	1.000	0.478
Albumin	(g/dL)	3.70 ± 0.480	3.77 ± 0.595	0.802	0.752
Creatinine	(mg/dL)	0.936 ± 0.302	1.10 ± 0.591	0.469	0.399
Separation of sequestrum	(−)	7	34	1.000	0.114
	(+)	5	7	3.469	
Osteolytic lesion	localized	7	27	1.000	0.735
	extended	5	14	1.378	
Periosteal reaction	(−)	11	23	1.000	0.038
	(+)	1	18	0.116	

## Data Availability

The datasets used and analyzed during the study are available from the corresponding author upon reasonable request.

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
