# Peer review of "Factors Exacerbating Clinical Symptoms and CT Findings in Patients with Medication-Related Osteonecrosis of the Jaw Receiving Conservative Therapy: A Multicenter Retrospective Study of 53 Cases"

_ijerph, 2022, doi:10.3390/ijerph19137854_

Round 1
Reviewer 1 Report
This is an interesting study with important findings. However, some concerns were raised.
1. There were 70 or 53 cases in this study? Please check the title.
2. Table. 1, It should be tested if there is a significant change between groups to confirm a good baseline for further investigation
3. Table.3 It should be tested the Odd ratio.
4. Data of this study were very interesting and important with many information. However, the analysis was quite simple and old. It can lead to false-positive (but there was no adjustment). It did also not take advance of those data, leading to superficial findings.
The authors have both quantitive, and qualitative variables, follow-up time, and observation (days). I suggest using a mixed-effect model or principal component analysis to mine the data and investigate the rate of treatment success.
Author Response
To reviewer #1
- There were 70 or 53 cases in this study? Please check the title.
(Reply)
The number of cases in the title was wrong. The title was corrected to “Factors exacerbating clinical symptoms and CT findings in patients with medication-related osteonecrosis of the jaw receiving conservative therapy: A multicenter retrospective study of 53 cases”.
- 1, It should be tested if there is a significant change between groups to confirm a good baseline for further investigation.
(Reply) Table 1 shows the patient's background factors. This study is a retrospective observational study of patients with MRONJ who underwent conservative therapy and was not divided into groups. Tables 3 and 4 show the differences between the factors in patients who improved clinically and those who did not, or those who did not change on the CT image and those who did worse.
- It should be tested the Odd ratio.
(Replay)
Odds ratios have been added to Table 3.
- Data of this study were very interesting and important with many information. However, the analysis was quite simple and old. It can lead to false-positive (but there was no adjustment). It did also not take advance of those data, leading to superficial findings. The authors have both quantitive, and qualitative variables, follow-up time, and observation (days). I suggest using a mixed-effect model or principal component analysis to mine the data and investigate the rate of treatment success.
(Reply)
To analyze factors related to therapeutic effects, analysis is usually performed using a logistic regression model or a cox proportional hazard model. However, the number of events is small (clinical healing + improvement is 15 cases, image healing + improvement is 2 cases), and it is considered that it is not appropriate to perform multivariate analysis, so the only univariate analysis was done in this study. As the Reviewer pointed out, we would like to increase the number of facilities and cases in the future to investigate these issues.
- I'm sorry that Table 4 is wrong with a different table. I changed to the correct table.
Reviewer 2 Report
Thank you for the opportunity to read the manuscript: Factors exacerbating clinical symptoms and CT findings in patients with medication-related osteonecrosis of the jaw receiving conservative therapy: A multicenter retrospective study of 70 cases
I have a number of comments that I hope will be helpful:
- Titles in the publishing system and the manuscript itself differ in the number of patients. The number of patients in the text of the manuscript does not match the number of patients in the title of the manuscript.
- The sentence from lines 52-53 requires justification of such a statement with literature sources, and preferably a more precise explanation of the authors' intentions. In my opinion, even patients in poor local or general condition can often qualify for minimally invasive surgical procedures such as sequestrectomy with curettage, administration of blood products such as PRF and local flap plastics.
- You mention tooth extraction and curettage under local anesthesia as conservative treatment - in my opinion, these are surgical procedures. Of course, I understand that you want to present these methods as much less invasive than resection treatment. However, please refer to the sources so that your qualification is justified or consider adding a mention that it is minimally invasive surgery and you intentionally classify it as a conservative method - then explain what you count as a surgical method.
- In the introduction you refer to the AAOMS guidelines, but you do not mention the MRONJ (0-3) stages and the fact that different stages are an indication for different treatments. Later on, these stages appear in the results section, so I think that it is even more necessary to mention them in the introduction.
- Isn't it a simplification to say that surgical treatment is first-line treatment in your centers? Maybe in stages 0-1 you decide to go for conservative treatment more often? I guess it might be wiser to delete this sentence.
- If possible, specify the location of the lesions more than just by specifying whether it is the maxilla or the mandible.
- If possible, specify what medications the patients were taking (there are many types of bisphosphonates and they differ in their potential to cause MRONJ). Haven't you had MRONJ cases in your study group after drugs other than bisphosphonates and denosumab?
- You probably should delete the supplementary materials entry because you haven't mentioned any.
- Many of the sources you use are older than 5 years. try to replace them with newer ones if possible. There is no shortage of new articles on MRONJ, and the approach to treating this disease is changing quite dynamically.
Overall, I find the manuscript very interesting. I would be grateful for considering my suggestions.
Author Response
To Reviewer #2
- Titles in the publishing system and the manuscript itself differ in the number of patients. The number of patients in the text of the manuscript does not match the number of patients in the title of the manuscript.
(Reply)
The number of cases in the title was wrong. The title was corrected to “Factors exacerbating clinical symptoms and CT findings in patients with medication-related osteonecrosis of the jaw receiving conservative therapy: A multicenter retrospective study of 53 cases”.
- The sentence from lines 52-53 requires justification of such a statement with literature sources, and preferably a more precise explanation of the authors' intentions. In my opinion, even patients in poor local or general condition can often qualify for minimally invasive surgical procedures such as sequestrectomy with curettage, administration of blood products such as PRF and local flap plastics.
(Reply)
Line 56: “conservative therapy is preferred in such cases” was revised to “minimally invasive surgical procedures or conservative therapy is preferred in such cases”. Other adjuvant therapy such as HBO, ozone, LLLT, L-PRF, teriparatide, BMP-2, etc., are described in the Discussion.
- You mention tooth extraction and curettage under local anesthesia as conservative treatment - in my opinion, these are surgical procedures. Of course, I understand that you want to present these methods as much less invasive than resection treatment. However, please refer to the sources so that your qualification is justified or consider adding a mention that it is minimally invasive surgery and you intentionally classify it as a conservative method - then explain what you count as a surgical method.
(Reply)
Line 76-78: “The teeth in the necrotic bone with symptoms such as pain were extracted. The sequestrum was removed, and the bone surface was curetted under local anesthesia.” was revised to “Further, removal of mobile segments of bony sequestrum, and extraction of symptomatic teeth within exposed necrotic bone are also considered in those undergoing conservative treatment.”.
- In the introduction you refer to the AAOMS guidelines, but you do not mention the MRONJ (0-3) stages and the fact that different stages are an indication for different treatments. Later on, these stages appear in the results section, so I think that it is even more necessary to mention them in the introduction.
(Reply)
The description of staging and treatment, and the new AAOMS Position Paper 2022 were added.
Line 43-44: “Conservative therapy was recommended as –” was revised to “In most stage 1 and 2 patients, conservative therapy was recommended as –“.
Line 47-50: The sentence “The AAOMS Position Paper has been revised in 2022 [4], and some modifications were made to the treatment strategy. It describes that both conservative and surgical treatments are acceptable for all stages of MRONJ, depending on the patient's situation.” was added
- Isn't it a simplification to say that surgical treatment is first-line treatment in your centers? Maybe in stages 0-1 you decide to go for conservative treatment more often? I guess it might be wiser to delete this sentence.
(Reply)
For all stages of MRONJ (stage 0-3), surgical treatment is the first-line treatment in both hospitals.
Line 70: “all stage of” was inserted before “MRONJ”.
- If possible, specify the location of the lesions more than just by specifying whether it is the maxilla or the mandible.
(Reply)
The site of MRONJ (anterior/posterior/anteroposterior) has been added to Table 1. The variable of the site was also added for the relationship with the treatment outcomes in Tables 3 and 4.
- If possible, specify what medications the patients were taking (there are many types of bisphosphonates and they differ in their potential to cause MRONJ). Haven't you had MRONJ cases in your study group after drugs other than bisphosphonates and denosumab?
(Reply)
Types of antiresorptive agents were shown in Table 1. There were no cases of MRONJ due to drugs other than bisphosphonate and denosumab.
- You probably should delete the supplementary materials entry because you haven't mentioned any.
(Reply)
The supplementary materials entry was deleted.
- Many of the sources you use are older than 5 years. try to replace them with newer ones if possible. There is no shortage of new articles on MRONJ, and the approach to treating this disease is changing quite dynamically.
(Reply)
We quoted the AAOMS Position paper (2022 revised edition) and added a description.
- I'm sorry that Table 4 is wrong with a different table. I changed to the correct table.
Round 2
Reviewer 1 Report
I agree with the current revised manuscript. Thank you
Author Response
Thank you very much for your kind peer review.
Reviewer 2 Report
Thank you kindly for the opportunity to evaluate the revised version of the manuscript. The authors referred to all my comments, and in most cases I consider the corrections made to be relevant and sufficient. I would appreciate an improvement on a few points that may still be unclear to the reader:
1. In the methodology, you specify the variables, including the site where MRONJ was diagnosed. Unfortunately, the terms "anterior, posterior, or anteroposterior" are not obvious to me. Could you please specify the source by which you defined the boundaries of these segments? Consider defining the area more clearly, e.g. by using oral sextants or by describing the area as incisal, canine, premolar and molar.
2. In Tables 1, 3 and 4, the addition of the variable "Site" did not refine the location of the MRONJ as this variable was treated regardless of the jaw identification. I consider it necessary to define the location of the lesions precisely, eg using sextants or the terms "premolar region of the upper jaw, molar region of the lower jaw". The current data showing, for example, a posterior / anteroposterior site, without specifying which jaw it is, do not make much sense.
3. The addition of the AAOMS Position paper (2022) is a good start. I will be grateful if you continue to work on updating the references section in line with my Round 1 comment.
Author Response
- In the methodology, you specify the variables, including the site where MRONJ was diagnosed. Unfortunately, the terms "anterior, posterior, or anteroposterior" are not obvious to me. Could you please specify the source by which you defined the boundaries of these segments? Consider defining the area more clearly, e.g. by using oral sextants or by describing the area as incisal, canine, premolar, and molar.
Line 88-89: The sentence was added.
The site was defined as “anterior” from the incisor to the canine region, “posterior” from the premolar region, and “anteroposterior” including both.
- In Tables 1, 3 and 4, the addition of the variable "Site" did not refine the location of the MRONJ as this variable was treated regardless of the jaw identification. I consider it necessary to define the location of the lesions precisely, eg using sextants or the terms "premolar region of the upper jaw, molar region of the lower jaw". The current data showing, for example, a posterior/anteroposterior site, without specifying which jaw it is, do not make much sense.
Tables 1, 3, and 4 were revised to describe the site of occurrence in more detail. However, due to the small number of cases in each category, it was not possible to show p-values or odds ratios.
- The addition of the AAOMS Position paper (2022) is a good start. I will be grateful if you continue to work on updating the references section in line with my Round 1 comment.
Line 291-300: Two new references (#40 and 41) and the next sentences were added.
As mentioned above, conservative therapy does not have a high cure rate for MRONJ, therefore, prevention of MRONJ is important. It was recommended that patients receiving antiresorptive agent therapy avoid invasive dental procedures such as tooth extractions, and that antiresorptive agents be withdrawn before tooth extraction, which resulted in the preservation of the tooth with a source of infection. Recently, Soutome et al. reported a significantly higher incidence of subsequent MRONJ in the presence of teeth with periapical lesions larger than 3 mm, periodontal pockets larger than 4 mm, and local infection symptoms [40]. Since extraction itself is not a risk factor for MRONJ, but local infection present in the tooth to be extracted is a risk factor, tooth extraction of such teeth may lead to the prevention of MRONJ development [41].
